# Fatty Acid Sensing in the Gastrointestinal Tract of Rainbow Trout: Different to Mammalian Model?

**DOI:** 10.3390/ijms24054275

**Published:** 2023-02-21

**Authors:** Jessica Calo, Sara Comesaña, Ángel L. Alonso-Gómez, José L. Soengas, Ayelén M. Blanco

**Affiliations:** 1Centro de Investigación Mariña, Laboratorio de Fisioloxía Animal, Departamento de Bioloxía Funcional e Ciencias da Saúde, Facultade de Bioloxía, Universidade de Vigo, 36310 Vigo, Spain; 2Departamento de Genética, Fisiología y Microbiología, Unidad Docente de Fisiología Animal, Facultad de Biología, Universidad Complutense de Madrid, 28040 Madrid, Spain

**Keywords:** gut sensing, fatty acids, gut-brain axis, feed intake, fish

## Abstract

It is well established in mammals that the gastrointestinal tract (GIT) senses the luminal presence of nutrients and responds to such information by releasing signaling molecules that ultimately regulate feeding. However, gut nutrient sensing mechanisms are poorly known in fish. This research characterized fatty acid (FA) sensing mechanisms in the GIT of a fish species with great interest in aquaculture: the rainbow trout (*Oncorhynchus mykiss*). Main results showed that: (i) the trout GIT has mRNAs encoding numerous key FA transporters characterized in mammals (FA transporter CD36 -FAT/CD36-, FA transport protein 4 -FATP4-, and monocarboxylate transporter isoform-1 -MCT-1-) and receptors (several free FA receptor -Ffar- isoforms, and G protein-coupled receptors 84 and 119 -Gpr84 and Gpr119-), and (ii) intragastrically-administered FAs differing in their length and degree of unsaturation (i.e., medium-chain (octanoate), long-chain (oleate), long-chain polyunsaturated (α-linolenate), and short-chain (butyrate) FAs) exert a differential modulation of the gastrointestinal abundance of mRNAs encoding the identified transporters and receptors and intracellular signaling elements, as well as gastrointestinal appetite-regulatory hormone mRNAs and proteins. Together, results from this study offer the first set of evidence supporting the existence of FA sensing mechanisms n the fish GIT. Additionally, we detected several differences in FA sensing mechanisms of rainbow trout vs. mammals, which may suggest evolutionary divergence between fish and mammals.

## 1. Introduction

In mammals, there is unequivocal evidence that the gastrointestinal tract (GIT) is critically involved in the homeostatic control of feeding and energy balance through the so-called gut-brain axis [1]. For this, the GIT contains intestinal cells able to sense the presence of nutrients (carbohydrates, fatty acids/lipids, and amino acids/proteins) in the lumen by specific “taste” receptors or transporters and respond to such information by releasing signaling molecules [2]. While three types of intestinal cells (enterocytes, brush cells, and enteroendocrine cells (EECs)) have been associated with nutrient sensing, the main chemosensory cells within the GIT are EECs [3]. When EECs sense nutrients, multiple regulatory peptides, mainly ghrelin (GHRL), cholecystokinin (CCK), peptide tyrosine-tyrosine (PYY), and glucagon-like peptide-1 (GLP-1), are released. These peptides can act paracrinally on neighboring cells, but their main role is to serve as signaling molecules for gut-brain communication, which can take place either by transmission through the vagus nerve or systemic circulation [2,3,4]. Information derived from the GIT reaches the central nervous system (CNS), where it is integrated, ultimately resulting in changes in the production of key hypothalamic factors that govern food intake [5].

Over the last decades, the mechanisms underlying nutrient sensing in the GIT have become an area of increasing scientific interest, and, to date, several carbohydrate, fatty acid, and amino acid sensing systems have been described in the mammalian EECs [2,3,4]. For the purpose of the present research, only sensing mechanisms involving lipids/fatty acids will be further described. The GIT is exposed to high levels of lipids derived from diet (mainly triglycerides, TGs), which, after lipase digestion in the small intestine, are cleaved to release free fatty acids (FFAs) that are sensed by different G protein-coupled receptors (GPCRs). The main GPCRs sensing FFAs are referred to as free fatty acid receptors (FFARs), and they respond to FFAs depending on the length of their aliphatic chain [6]. Thus, medium-chain (6–12 carbons) and long-chain (13–21 carbons) fatty acids (MCFAs and LCFAs, respectively) are detected by FFAR1 (previously termed as GPR40) and FFAR4 (or GPR120), both primarily located in I- and L-cells [6,7,8]. In contrast, FFAR2 and FFAR3 (previously named GPR43 and GPR41, respectively) are responsive to short-chain fatty acids (SCFAs; <6 carbons) such as butyrate, propionate or acetate, which can be acquired from food but predominantly derive from the metabolism of non-digestible carbohydrates by gut microbiota in the distal intestine; because of this, FFAR2 and FFAR3 are expressed at large amounts in colonic L-cells [6,9]. Apart from these major receptors, GPR84 has been later discovered to bind MCFAs [10], although available evidence indicates that this receptor is not very abundant in the mammalian GIT and that it is not expressed in EECs. Thus, its role as a fatty acid sensor appears to be secondary; instead, its major role seems to be to enhance pro-inflammatory signaling and macrophage effector functions [11]. Finally, the receptor GPR119 has been considered an intestinal lipid sensor, although its natural ligands are not typically FFAs but endogenous lipid derivatives such as oleoylethanolamide (OEA) [12]. Nevertheless, a recent study has reported GPR119 activation in response to FFAs such as palmitoleic acid in human islet EndoC-betaH1 cells [13]. GPR 119 is also activated by dietary TG-derived 2-monoacylglycerols (2-MAG) [14]. Indeed, GPR119 is at least as important as FFAR1 in mediating the TG-induced secretion of gastrointestinal incretins in the small intestine, co-acting in synergy with FFAR1 [14]. Finally, in the colon, GPR119 is activated by microbiota-derived metabolites [15]. Besides GPCRs, several carriers have been associated with fatty acid sensing in the mammalian GIT. These include the fatty acid transporter CD36 (FAT/CD36), the fatty acid transport protein 4 (FATP4), and the monocarboxylate transporter isoform-1 (MCT-1) [2,3,4]. Fatty acid carriers are typically located on the apical membrane of enterocytes, where they facilitate FFA uptake. FAT/CD36 and FATP4 appear to be involved in LCFA translocation along the intestine, while MCT-1 participated in the absorption of SCFAs in the colon. MCFAs are absorbed by passive diffusion [2,4]. Despite its predominant enterocyte location, some studies have demonstrated the presence of some fatty acid transporters (at least FAT/CD36) in EECs, where they contribute to lipid-derived gastrointestinal hormonal release [16].

Gut nutrient sensing mechanisms remain almost unexplored in fish. In a previous recent study from our research group, we identified that the rainbow trout (*Oncorhynchus mykiss*) genome contains 10 different isoforms of *ffar* genes, of which only *ffar1* seems clearly homologous to its mammalian counterpart. By contrast, the remaining isoforms identified appear to have evolved independently and it is not clear whether they are homologous to mammalian genes. In addition, we observed that a gene encoding a Ffar4 receptor subtype is missing in the rainbow trout. These observations allow us to suggest functional differences in gut fatty acid sensing between mammals and rainbow trout. However, as far as we are aware, there is no information in the literature regarding gut fatty acid sensing mechanisms in fish. Besides all the general roles of lipids in vertebrates [17], this nutrient type is particularly relevant for fish because the major aerobic fuel source for energy metabolism of fish muscle is FFAs derived from triglycerides (as those in diet) [18], and the main source of energy in aquaculture nutrition is lipids [19]. Additionally, it is important to note that fish and mammals differ importantly in terms of lipid metabolism (e.g., fish have the ability to produce long-chain polyunsaturated fatty acids (PUFAs), essential for multiple physiological processes, endogenously, while most mammals have a very low capacity for PUFA synthesis [20]), thus being of enormous interest to study whether evolutionary variations in terms of sensing mechanisms may exist between two groups derived by such differences. With this background, the present study aimed to identify and characterize fatty acid sensing mechanisms in the GIT of a fish model with a great interest in aquaculture, the rainbow trout. These carnivorous species have a better ability to digest lipids compared to herbivorous and omnivorous species, which appears to be attributed to their more specific and higher lipase activity and/or their genetic potential to store lipids [21]. Therefore, the comparison between rainbow trout and the known mammalian models provide two different frames, mostly unknown: evolutionary trends within vertebrates and putative differences between carnivore species and the omnivore models assessed so far in mammals.

## 2. Results

### 2.1. Fatty Acid Receptors and Transporters mRNAs Are Differentially Expressed along the Rainbow Trout Gastrointestinal Tract

As shown in Figure 1, mRNAs encoding different Ffar isoforms, Gpr84, Gpr119, Fat/cd36, Fatp4, and Mct-1 (a/b), are found, at different abundance levels, in almost all of the regions of the rainbow trout GIT studied, i.e., stomach, pyloric caeca, proximal intestine, middle intestine, and distal intestine. Specifically, *ffar1* mRNAs were more abundantly expressed in the pyloric caeca and proximal intestine, followed by the rest of the intestinal sections, with undetected expression in the stomach (Figure 1A). The abundance of *ffar2b1.1* mRNAs was higher in the pyloric caeca compared with the rest of the tissues analyzed, but quantifiable levels were also observed in the proximal and middle intestine; however, expression levels were extremely low in the stomach and hindgut, thus hampering gene expression quantification in these regions (Figure 1B). *ffar2b1.2* mRNAs were more abundant in the pyloric caeca, followed by the proximal and distal intestine, although low levels were detected in all gastrointestinal regions (Figure 1C). mRNAs encoding Ffar2b2a and Ffar2b2b were the most abundant of all receptor mRNAs studied (Ct values ≈28) and were detected along the entire GIT, with the highest levels found in distal intestine (Figure 1D,E). *ffar2a1b* mRNAs were higher in the intestine compared to the stomach, and pyloric caeca, with the highest levels detected in the proximal and distal regions, (Figure 1F). The expression of *ffar2a2* mRNAs was high in the distal intestine, low in the rest of the intestinal regions, and almost undetected in the stomach (Figure 1G). Both *gpr84* and *gpr119* mRNAs were predominantly found in the distal intestine, although quantifiable expression levels were observed in all gastrointestinal regions (Figure 1H,I). It should be noted that the abundance of all receptors mRNAs throughout the gastrointestinal, yet quantifiable, was rather low, as indicated by high Ct values in the real-time PCR runs (≈28–34). In contrast, lower Ct values (≈25–29), and therefore greater expression levels, were observed for mRNAs encoding the fatty acid transporters Fat/cd36, Fatp4, and Mct-1a. All three were abundantly expressed throughout the entire GIT, although some differences in expression levels were detected among regions for *fatp4* (lower relative expression in the stomach) and *slc16a1a* (higher relative expression in the distal intestine) (Figure 1J–M). The expression of *slc16a1b* was high in the stomach, with levels compared to the rest of transporters throughout the GIT (Ct values ≈25), but very low in pyloric caeca and all intestinal regions (Ct values ≈30–31) (Figure 1M).

### 2.2. Luminal Fatty Acids Modulate Fatty Acid Receptors and Transporters mRNA Expression in the Gastrointestinal Tract

Fish fasted for 48 h were intragastrically administered with octanoate, oleate, ALA, or butyrate, and samples of different regions of the GIT were collected at 20 min and 2 h post-administration to assess different parameters related to fatty acid sensing and appetite regulation (Figure 2A). Figure 2B–M shows the effects of intragastrically administered fatty acids on the mRNA expression of fatty acid receptors and transporters along the rainbow trout GIT at 20 min and 2 h post-administration. In a short time, treatment with octanoate led to a significant upregulation of *ffar1* and *ffar2b1.2* in the middle intestine (Figure 2B,D), *ffar2a1b* in the proximal and middle intestines (Figure 2G), *ffar2a2* in the stomach and middle intestine (Figure 2H), and *slc16a1a* (encoding Mct-1a) in the middle intestine (Figure 1M). Oleate induced *ffar2b1.1*, *ffar2b1.2*, *ffar2a1b*, *ffar2a2*, *gpr84* and *gpr119* in proximal intestine (Figure 2C,D,G–J), and also increased *ffar1* in middle intestine (Figure 2B), and *ffar2b1.2* in stomach (Figure 2D), while it decreased *gpr84* in the distal intestine (Figure 2I) and *fatp4* in the proximal intestine (Figure 2L). In addition, both octanoate and oleate significantly increased *gpr119*, *cd36*, and *fatp4* mRNAs in the distal intestine (Figure 2J–L). Administration of ALA resulted in increased levels of *ffar2b1.1* and *ffar2b2a* in the proximal intestine (Figure 2C,E), *ffar1* and *ffar2b1.2* in the proximal and middle intestines (Figure 2B,D), *cd36* and *fatp4* in the distal intestine (Figure 2K,L), and *gpr119* in all regions of the GIT analyzed, except for the stomach (Figure 2J). On the contrary, significant ALA-induced downregulations of stomach *slc16a1a* (Figure 2M) and distal intestine *ffar2a1b* and *gpr84* (Figure 2G,I) was detected. Finally, significantly higher levels of *gpr84* and *gpr119* in the stomach (Figure 2I,J), *ffar2a1b*, *gpr84*, *cd36*, *fatp4*, and *slc16a1a* in the proximal intestine (Figure 2G,I,K–M), *gpr84*, *gpr119*, and *cd36* in the middle intestine (Figure 2I–K), and *ffar2b2b* and *slc16a1a* in the distal intestine (Figure 2F,M) were observed in fish administered with butyrate compared to control fish. Butyrate treatment also led to decreased *slc16a1b* mRNAs in the stomach (Figure 2N), *ffar2b1.1* in the middle intestine (Figure 2C), and *fatp4* mRNAs in the stomach (Figure 2L).

At 2 h post-administration, a significant increase in the mRNA levels of *ffar2b2a*, *ffar2a1b,* and *gpr84* in the stomach (Figure 2E,G,I), *cd36* in the proximal and middle intestine Figure 2K, and *fatp4* in the stomach and proximal intestine (Figure 2L), was observed in response to octanoate. Oleate up-regulated the expression of *ffar2b2a*, *gpr84,* and *cd36* in the stomach (Figure 2E,I,K), of *ffar1* in the proximal intestine (Figure 2B), and of *ffar1*, *ffar2b1.2*, and *gpr84* in the distal intestine (Figure 2B,D,I). On the contrary, it down-regulated the mRNA abundance of *gpr119* in the distal intestine (Figure 2J). Treatment with ALA resulted in significantly higher levels of *ffar1* and *ffar2b2a* in the stomach (Figure 2B), *ffar2b1.2* in the proximal, middle, and distal intestine (Figure 2D), *ffar2a1b* in the distal intestine (Figure 2G), *gpr84* in the stomach, proximal intestine, and middle intestine (Figure 2I), *fatp4* in the stomach and proximal intestine (Figure 2L), and *slc16a1a* in the distal intestine (Figure 2M). Lastly, increased levels of *ffar2b1.2* and *ffar2b2b* in the middle and distal intestine (Figure 2D,F), *ffar2a1b* in the stomach and proximal intestine (Figure 2G), *ffar2a2* and *slc16a1a* in proximal intestine (Figure 2H,M), *gpr119* in the distal intestine (Figure 2J), *fatp4* and *slc16a1b* in the proximal and middle intestine (Figure 2L,N), and *cd36* in all gastrointestinal regions analyzed (Figure 2K), were detected upon butyrate administration.

### 2.3. Gastrointestinal mRNA Expression of Intracellular Signaling Molecules Is Altered by the Luminal Presence of Fatty Acids

Considering that gustducin is the major G protein activated in response to FFAR activation in the mammalian GIT [2] and that the phospholipase C (PLC)- inositol triphosphate (IP_3_) and adenylate cyclase (AC)-cAMP-protein kinase A (PKA) pathways are the main intracellular signaling cascades triggered as a consequence [3,4,5,6], in this study we measured the gastrointestinal mRNA expression of a putative G protein involved in nutrient signaling in fish (Gnai1) as well as the mRNA levels of key elements of both the PLC-IP3 and AC-cAMP-PKA pathways in response to fatty acid administration to study whether the same mechanisms may operate in rainbow trout. The changes in mRNA abundance of such parameters at 20 min and 2 h after intragastric administration of fatty acids are shown in Figure 3 and Appendix A. Increased *gnai1* mRNAs were observed upon octanoate treatment at 20 min in the middle and distal intestines, upon oleate treatment at 2 h in proximal and middle intestines, upon ALA treatment at 20 min in proximal and distal intestines, and at 2 h in the stomach and all intestinal regions, and upon butyrate treatment at 20 min in the stomach. Expression of *plcβ1* was found to be up-regulated by butyrate in the stomach, proximal and middle intestines at 20 min, while at 2 h, only a significant upregulation was detected in the stomach. Oleate and ALA also caused significant increases in *plcβ1*, as well as *plcβ3*, mRNAs, especially in the proximal and middle intestines. On the contrary, the expression of both genes remained unaltered or even down-regulated in response to octanoate. Except for the middle intestine, all fatty acids tested led to significantly lower levels of *plcβ4* mRNAs compared to the control group in all or most regions tested and at 20 min and/or 2 h. As for *itpr1*, we found increased mRNA levels at 20 min in the proximal intestine in response to butyrate, in the middle intestine in response to all fatty acids, and in the distal intestine in response to octanoate and ALA, as well as at 2 h in the proximal intestine in response to ALA and butyrate, middle intestine in response to ALA and distal intestine in response to butyrate. The expression of *itpr3* was, in general, down-regulated in response to the luminal presence of fatty acids, with the major changes found in the stomach. Finally, *ac* mRNA levels were observed to be downregulated by butyrate treatment in the proximal intestine at 20 min and in the middle intestine at 20 min and 2 h. However, increased *ac* mRNAs were found in the middle intestine 2 h after oleate administration and in the distal intestine 20 min after ALA administration.

### 2.4. Abundance of Gastrointestinal Hormones Responds to Luminal Fatty Acids

The luminal presence of fatty acids modulates mRNA and protein levels of gastrointestinal hormones, as shown in Figure 4. The abundance of *ghrl* mRNAs was observed to be up-regulated by ALA and butyrate in the stomach, proximal intestine (only butyrate), and middle intestine at 20 min post-treatment and by oleate in the stomach and ALA in the proximal and middle intestine at 2 h. No significant differences in stomach Ghrl levels were detected in response to any of the fatty acids (Figure 4B,H). Levels of *cck*/Cck were unaltered by luminal fatty acids at 20 min (Figure 4C,D,H). At 2 h, oleate and ALA led to a significant upregulation of *cck* and/or Cck levels in the proximal intestine, while the opposite effect was observed for butyrate (Figure 4C,D,H). In addition, oleate and butyrate led to increased *cck* mRNAs in the distal intestine (Figure 4C). Treatment with octanoate, oleate, and ALA resulted in a general tendency to increase the abundance of *pyy*/Pyy, especially at 2 h (Figure 4E,F,H). Butyrate, on the other hand, reduced *pyy* mRNA levels in both the proximal and middle intestine at 20 min and 2 h (Figure 4E), although a significant increase in Pyy protein levels was observed in the proximal intestine at 2 h (Figure 4F,H). Finally, *gcg* (*proglucagon*, gene encoding Glp-1) mRNAs were found to be up-regulated by octanoate, oleate, and ALA treatment at 20 min in the proximal intestine, while down-regulated by the former two at the same time point in the distal intestine (Figure 4G). Due to technical difficulties with finding a suitable Glp-1 antibody, we were not able to detect levels of this protein by Western blot in the present study.

## 3. Discussion

Great interest in elucidating the mechanisms by which the gut senses luminal nutrients and how this sensing impacts the homeostatic control of feeding has been taking place over the last few years. Gut nutrient sensing relies on the presence of specific receptors and transporters located mainly in the luminal surface of enteroendocrine cells and enterocytes, respectively, which are able to respond to variations in the luminal levels of nutrients [7,8,9]. Studies on this topic, however, have focused mainly on mammalian models, and whether equivalent mechanisms operate in other vertebrate groups remains practically unknown, particularly in fish. This research aims to address this scarcity of information in the fish literature and provides the first evidence on the presence and functioning of fatty acid sensing mechanisms in the GIT of a carnivore fish species, the rainbow trout. We focused on lipids because of three reasons: (i) they are the main energy source in aquaculture nutrition [11], (ii) they are the major aerobic fuel source for energy metabolism of the fish muscle [10], and (iii) there are several key differences between fish and mammalian lipid metabolism [12].

### 3.1. Fatty Acid Transporters and Carriers in Rainbow Trout GIT and Their Involvement in Sensing Different Types of FAs

The first objective of our research was to study whether the main fatty acid receptors and transporters described in the mammalian GIT (i.e., FFAR1/2/3/4, GPR84/119, FAT/CD36, FATP4 and MCT-1; [7,8,9]) are present in rainbow trout. A previous *in silico* study from our research group, together with a study by Roy and coworkers (Roy et al., under review), described that some genes encoding such mammalian receptors (particularly Ffars) are not present within the rainbow trout genome (*ffar4*), some are duplicated (as expected due to whole-genome duplications events during evolution), and some appear not to be orthologous to mammalians. In the same previous study, we showed that most of these *ffar* genes identified within the rainbow trout genome (specifically, *ffar1*, *ffar2b1.1*, *ffar2b1.2*, *ffar2b2a*, *ffar2b2b*, *ffar2a1b,* and *ffar2a2*) are expressed at a smaller or greater extent in the stomach, anterior intestine and/or posterior intestine. In the present study, we carried out PCRs targeting these genes to confirm previous observations. However, the putative presence of the rest of the fatty acid receptors and fatty transporters remains unknown. Fatty acid transporter genes are pretty well conserved throughout evolution, and sequences encoding Fat/cd36, Fatp4, and Mct-1 (with two copies for the latter) can all be found within the rainbow trout genome [14,15]. A high degree of conservation is also observed for genes encoding Gpr84 and Gpr119 [16]; thus, their presence in the rainbow trout genome is also evident. PCR and RT-qPCR analyses targeting all these genes indicated the expression of mRNAs encoding Gpr84, Gpr119, Fat/cd36, Fatp4, and Mct-1 (a and b isoforms) in the GIT of rainbow trout. In the mammalian gut, fatty acid receptors (except for GPR84, [17]) and transporters are located in the apical membrane of different cell types, with receptors being typically found in enteroendocrine cells while transporters in enterocytes [7,8,9]. Experimental approaches used in the present research do not allow us to discriminate the cell type location of receptors or transporters, so we will discuss obtained results considering gastrointestinal cells in general. However, the fact that the mRNA abundance of receptors was low (very high Ct values) and that of transporters considerably high might be an indirect indicator of their cell type location. Thus, considering that enterocytes are the most abundant epithelial cells in the GIT, and EECs represent only 1% of them [18], we might suggest receptor presence in EECs and transporter in enterocytes. Future lines of research will focus on the sorting of rainbow trout intestinal epithelial cells by type using flow cytometry and the study of nutrient-sensing mechanisms taking place in each individual cell type.

Based on the GIT distribution study, we observed that except for *ffar1* and *ffar2b1.1,* whose transcripts were not quantifiable in the stomach, and also hindgut in the case of the latter, all fatty acid receptors and transporters detected are found at quantifiable levels in the stomach, pyloric caeca, and along the entire intestine, thus showing a widespread distribution within the GIT. This differs from the mammalian model for some receptors/transporters. For instance, any receptor was observed to be almost exclusively expressed in the distal intestine of the rainbow trout, as FFAR2 and FFAR3 are in the case of mammals [19,20]. It is also worth pointing out the case of MCT-1. Two isoforms of this transporter (a and b) have been described in rainbow trout [21]. These forms were here observed to show a very different expression profile along the GIT, with *slc16a1a* (encoding Mct-1a) mRNAs expressed in the whole GIT but most importantly in the hindgut, and *slc16a1b* (encoding Mct-1b) almost exclusively detected in the stomach. In mammals, studies in mice and rats have shown that the single MCT-1 isoform found in these vertebrates is poorly expressed in the stomach but abundant in the colon [22], as expected considering that MCT-1 is involved in SCFA uptake and that the colon is the predominant location for SCFA synthesis. However, interestingly, a high expression of this transporter was reported in both the caprine stomach and large intestine [23], which appears to be related to the fact that ruminants also produce large amounts of SCFAs in the rumen. Indeed, SCFAs constitute the major fuel source in ruminants, providing up to 80% of their energy requirements [24]. The physiological significance of the distinct expression profile of the two Mct-1 isoforms along the rainbow trout GIT observed in this study requires additional investigation. However, we could hypothesize that each isoform, predominant at each end of the GIT, might be involved in the uptake of different SCFAs. This could relate to the previous report that the microbiota of the rainbow trout stomach and intestine shows considerable differences [25].

The next step of our study was to characterize the response of the identified receptors to the luminal presence of fatty acids of different lengths and degrees of unsaturation [i.e., octanoate (8-carbon saturated FA), oleate (18-carbon monounsaturated FA), ALA (18-carbon PUFA), and butyrate (4-carbon saturated FA)]. For this, we intragastrically administered fatty acids into fasted rainbow trout and assessed the abundance of mRNA encoding the target receptors in the GIT. The most important changes include increases in the mRNA abundance of *ffar2a1b* and *ffar2a2* in response to octanoate, *ffar1*, *ffar2b1.1* and *ffar2b1.2* in response to oleate, *ffar1*, *ffar2b1.2* and *gpr119* in response to ALA, and *ffar2b2b,* and *gpr84* in response to butyrate, particularly in anterior regions of the GIT (importantly involved in nutrient sensing in mammals). Some of the fatty acids tested, mainly oleate and ALA, also led to increased expression of other types of fatty acid receptors (e.g., ALA up-regulated the expression of *ffar2b1.1*, *ffar2a* and *gpr119*, while oleate that of the *ffar2a*); however, these increases were, in general, less pronounced than the formers. This suggests that the different fatty acid receptors appear to be more responsive to specific ligand/s (i.e., Ffar1:oleate and ALA, Ffar2b1 (1 and 2): oleate and ALA, Ffar2b2b: butyrate, Ffar2a1b: octanoate and butyrate, Ffar2a2: octanoate, Gpr84: butyrate, Gpr119: ALA), although they may also be activated by other types of fatty acids. While we only measured mRNA abundance here, and experiments testing the ligand affinity of each receptor are required, the activation profile of fatty acid receptors in the rainbow trout GIT that can be suggested from our experiment points out important putative differences with regard to gut fatty acid sensing mechanisms in mammals, in which FFAR1 is only activated by MCFAs and LCFAs, FFAR2 and FFAR3 are only activated by SCFAs, GPR84 mainly by MCFAs, GPR119 by lipid derivatives (e.g., OEA) [7,8,9]. These differences strengthen our hypothesis of rainbow trout not having clear orthologous receptors to mammalian FFFAR2 and FFAR3. Although a deeper understanding of the mechanisms underlying fatty acid sensing in the rainbow trout GIT is required, present observations establish a basis in favor of the existence of major functional (maybe evolutionary) differences between gut nutrient sensing mechanisms between fish and mammals.

An interesting observation from our intragastric experiment is that there is a clear differentiation in the activation of receptor mRNA abundance in response to the luminal presence of fatty acids depending on the region of the GIT and time. Such a differentiation also applies to the mRNA abundance of the fatty acid transporters tested, i.e., Fat/cd36, Fatp4, and Mct-1a/b (Figure 5). In general terms, we can distinguish between one type of response in the anterior region of the GIT (including stomach, proximal intestine, and likely middle intestine) and another type of response in the distal intestine. In addition, results obtained point towards comparable mechanisms of action for octanoate, oleate, and ALA, while butyrate displayed clear differences. With this in mind, major results from our study allow us to hypothesize that the presence of octanoate, oleate, ALA, or butyrate in the intestinal lumen would be first sensed by specific membrane receptors located in the anterior regions of the GIT. As mentioned earlier, although functional studies on ligand affinity are needed, we propose that Ffar2a (1b and 2) could be more activated by octanoate, Ffar1 and Ffar2b1 (1 and 2) by oleate, Ffar1, Ffar2b1 (1 and 2) and Gpr119 by ALA, and Ffar2a1b and Gpr84 by butyrate, although less pronounced activations with other fatty acids may occur. Besides receptors, the butyrate-induced increased expression of *cd36*, *fatp4* and *slc16a1a* in the proximal intestine suggests the transporters Fat/cd36, Fat/p4, and Mct-1a as additional important sensors of butyrate in the rainbow trout GIT at a short-time. This observation differs from the mammalian model, in which only MCT-1, and not FAT/CD36 or FATP4, plays a role in the intestinal transport of SCFAs like butyrate [26]. Interestingly, our results demonstrated the increased abundance of mRNAs encoding Mct-1a not only in response to butyrate but also to octanoate in the proximal and middle intestine, which points towards this fatty acid as an additional activator of Mct-1a in the rainbow trout GIT. Except for this, treatment with octanoate, oleate, and ALA led to a general inhibition of the transporter’s mRNA abundance in the stomach and proximal intestine. Since we measured mRNA abundance only, this downregulation does not discard Fat/cd36, Fatp4, and Mct-1a/b as putative sensors for octanoate, oleate, and ALA, but could be the result of another response (e.g., negative feedback), although further studies are required for a certain explanation.

In the distal intestine, unlike what has just been stated, we observed that octanoate, oleate, and ALA increased *cd36* and *fatp4* mRNA abundance. This response can be attributed to the fact that the number of fatty acids in the distal vs. proximal intestine is likely considerably lower and/or that the mRNA levels of both *cd36* and *fatp4* are lower in the distal vs. proximal intestine/stomach (as observed from the GIT distribution study). Therefore, an increase in transporter expression in response to fatty acids may be related to transporter sensitivity increase. In any case, all three fatty acids (not only oleate) increased *cd36* and *fatp4* mRNA abundance (and considering that this observation might be an indicator of increased transport activity) is different from the mammalian model, in which both FAT/CD36 and FATP4 are in charge of LCFA translocation, whereas MCFAs are absorbed by passive diffusion [7,9]. However, again, this observation is just based on mRNA abundance data, and further research devoted to the study of transporter activity in response to different fatty acids is needed to confirm that both FAT/CD36 and FATP4 would be translocating fatty acids of different lengths (LCFAs, MCFAs, and PUFAs) in the rainbow trout distal intestine. The translocation model for SCFAs would likely occur according to the mammalian model [26], with MCT-1 (specifically, Mct-1a isoform in rainbow trout) being responsible for such an action, as suggested for the increased *slc16a1a* mRNA abundance and unaltered/decreased *cd36* and *fatp4* mRNA abundance in the distal intestine in response to butyrate. As for the receptors, Gpr119 and Ffar2b2b appear to be the only receptor types detecting the luminal presence of fatty acids (octanoate, oleate, and ALA in the case of the former, and butyrate in the latter) in the distal intestine at a short-term, as depicted by increased *gpr119* (and not other receptors) expression in response to octanoate, oleate, and ALA, and increased *ffar2b2b* in response to butyrate.

Over a long-time, major results demonstrated that all fatty acids up-regulated the mRNA abundance of *cd36* and/or *fatp4* in proximal areas of the GIT. This result seems controversial when compared with the octanoate/oleate/ALA-induced down-regulation of the mRNA abundance of both transporters in the proximal intestine at 20 min. However, as discussed for the distal intestine, such up-regulation after 2 h could increase the sensitivity of the transporters in response to low luminal levels of fatty acids (as there would likely be compared to 20 min). In any case, these results support the wider affinity of FAT/CD36 and FATP4 to fatty acids of different lengths in rainbow trout vs. the restricted affinity in mammals [7,9]. In contrast, the induced expression of *slc16a1a* and *slc16a1b* in the proximal and/or middle intestine in response to butyrate (and not to other fatty acids) argues in favor of Mct-1 being more devoted to the translocation of SCFAs rather to other fatty acid types. Unlike proximal gastrointestinal regions, no major fatty acid-induced changes in the transporter mRNA abundance (except for a butyrate-induced increase in *cd36* mRNAs) were detected in the distal intestine, which may indicate that transport of at least octanoate, oleate, and ALA into distal intestinal cells occur at a shorter time. Regarding receptors, we observed a general attenuation in expression activation of fatty acid receptors mRNA abundance compared to 20 min, as observed, for instance, in the cases of *ffar2b1.1* (unaltered upon all treatments) and *gpr119* (only activated in response to butyrate in the distal intestine). Other receptors, such as *ffar1*, showed a similar induction in expression to that observed at 20 min, i.e., mainly in response to oleate and ALA in the proximal and middle intestine. Expression of *ffar2b1.2* was also mainly induced by the same ligands (oleate and ALA) but at more distal regions of the GIT. Finally, we can highlight the case of *gpr84*, which was observed to be increased in the stomach 2 h after intragastric fatty acid administration regardless of the fatty acid assessed. It might be possible that these observations respond to an increase in receptor sensitivity over time. Altogether, results from the present research clearly suggest a differential activation profile of fatty acid receptors along the rainbow trout GIT depending on the time after nutrient administration.

### 3.2. Intracellular Mechanisms Triggered and Hormone Release as a Consequence of Gastrointestinal Fatty Acid Receptor Activation

Fatty acid receptors, as classical GPRs, respond to fatty acid binding with structural changes that lead to the activation of intracellular guanine nucleotide-binding proteins (G proteins) and the subsequent triggering of diverse signaling pathways. The major G protein coupling FFAR activation to hormonal release in the mammalian GIT appears to be gustducin (initially found in taste cells) [2]. Nevertheless, it appears that no ortholog of the mammalian gustducin gene (*gnat3*, guanine nucleotide-binding protein g (t) subunit alpha-3) is present in teleost fish; instead, other G proteins (e.g., Gnai1) appear to participate in the signaling of gut sensing [27,28]. The general up-regulation of *gnai1* mRNA abundance in response to intragastrically administered fatty acids observed in this study argues in favor of this G protein being activated as a consequence of fatty acid binding to FFARs in the rainbow trout GIT. In mammals, different signaling pathways appear to be triggered upon G protein activation depending on the receptor. Thus, the major effector for FFAR2 and FFAR3 seems to be PLC, whose activation results in increased production of IP_3_, which in turn binds to its receptor (ITPR3) located at the endoplasmic reticulum, releasing Ca^2+^ into the cytoplasm [3]. GPR119 operates mainly through the AC-Camp-PKA pathway: its activation results in the activation of AC, responsible for converting ATP to the second messenger Camp, thus leading to Camp accumulation and, thereby, activation of PKA [4]. In the case of GPR84, signaling pathways downstream of its activation have been well studied regarding its pro-inflammatory nature. Considering that elevated intracellular Camp levels suppress innate immune functions, it has been proposed that GPR84 exerts its pro-inflammatory actions by inhibiting AC and thereby suppressing intracellular Camp [5,6]. Additionally, other signaling pathways, such as the ERK cascade, have been associated with GPR84 signaling in immune functions [29]. Nevertheless, no information is available on the specific intracellular cascades in charge of coupling GPR84 and gastrointestinal hormone release. With this background, and considering that no evidence in this respect is available in fish literature, we investigated in the present study whether the gastrointestinal abundance of mRNAs encoding key elements within the PLC-IP_3_ and AC-Camp-PKA pathways is affected by the luminal presence of fatty acids. The results demonstrated increased levels of *plcβ1*, *plcβ3,* and *itpr1* mRNAs in anterior regions of the rainbow trout GIT in response to oleate and ALA, which may indicate that these two fatty acids could possibly signal through the PLC-IP_3_ pathway, although some important differences, such as the involvement of the Plcβ1 and 3 (instead of PLCβ2) and Itpr1 (instead of ITPR3), may exist with respect to the mammalian model. The PLC-IP_3_ pathway (specifically involving the isoforms Plcβ1 and Itpr1) may also participate in mediating butyrate actions in anterior regions of the rainbow trout GIT. As for the AC-cAMP-PKA signaling cascade, it might mediate at least some ALA responses in the distal intestine, maybe by acting through GPR119, as suggested by increased *ac* mRNA levels upon treatment with this fatty acid in the mentioned region. We also observed an interesting down-regulation of *ac* mRNAs in the proximal and middle intestine upon butyrate treatment, which, considering the role herein proposed for GPR84 in the mediation of butyrate responses, may match the intracellular signaling cascade proposed to this receptor in mammals [5,6]. Notably, no major changes occurred in the mRNA abundance of the intracellular signaling elements tested in response to octanoate, suggesting that other signaling pathways different from PLC-IP_3_ and AC-cAMP-PKA likely mediate octanoate actions. It has to be taken into consideration, however, that all these observations are based on gene expression data only, and future studies measuring the levels of second messengers should be performed in order to confirm present results.

In mammals, the triggering of intracellular signaling cascades in response to the sensing of luminal fatty acids and leads to the release of gastrointestinal hormones [8,9]. In the case of FFAR2 and 3, such a release occurs as a consequence of the rise in intracellular Ca^2+^, which activates the fusion machinery of the secretory granules containing hormones, thus triggering their release by exocytosis. Ca^2+^-triggered exocytosis likely operates for GPR119 as well, with PKA acting as a regulator of such a process [30]. Experimental approaches included in this study do not allow to describe the triggering process underlying hormonal release, but both qPCR and Western blot analysis demonstrated increased mRNA/protein levels of major gastrointestinal hormones (Ghrl, Cck, Pyy, and/or Glp-1) in the rainbow trout GIT upon fatty acid intragastric treatment, suggesting that hormone release is a consequence of gut fatty acid sensing in rainbow trout, as is the case in mammals [8,9]. Major increases in gastrointestinal hormone levels occurred in anterior regions of the GIT (stomach, proximal, and, to a lesser extent, middle intestine), suggesting these regions are primarily involved in appetite regulation. We observed a differential modulation of the Ghrl, Cck, and Pyy mRNA and/or protein level abundance depending on the fatty acid. In general, both octanoate and oleate led to increased Glp-1 levels at a short time-post administration, while at a longer time, they led to increased Pyy and also Cck in the case of oleate. All these hormones are of anorexigenic nature [31,32]; thus, their release in response to octanoate and oleate would be in agreement with an inhibitory role in feed intake for these two fatty acids. In the case of oleate, present observations regarding the hormonal release are in accordance with mammalian studies, which reported that LCFAs trigger CCK, GLP-1, and PYY secretion and suppress ghrelin release [33,34]. However, this response was not seen with fatty acids of 11 carbon atoms or fewer [35]; thus, results here observed for octanoate support a different model than that known in mammals. In mammals, both octanoate and oleate are primarily sensed by FFAR1 and FFAR4, and these two receptors are, therefore, related to hormone release. MCFAs are also detected by GPR84, but this receptor in mammals appears not to be expressed in EECs; thus, it would not be involved in the release of gastrointestinal hormones. In rainbow trout, we proposed that Ffar4 is absent, and thus the receptor binding these two fatty acids in this species (apparently n-Ffar5b (1b and 2a) in the case of octanoate and Ffar1 and n-Ffar2a (1 and 2) in the case of oleate) would be associated with octanoate- and oleate-evoked hormone release. FFAR2 and FFAR3 in mammals are responsive to SCFAs (such as butyrate), and they are believed to induce the release of PYY [36,37] and GLP-1 [38]. However, a later study using isolated rat colons suggested that the release of colonic PYY/GLP-1 in response to the presence of luminal SCFAs does not involve FFAR2/FFAR3; it rather occurs in response to the metabolization of SCFAs and subsequent function as a colonocyte energy source [39]. Results from the present study using butyrate demonstrated, in general lines, increased levels of Ghrl and a decrease in those of Cck and Pyy in response to this SCFA, hormonal responses that would presumably occur upon activation of n-FFar2b2b, n-Ffar5b1b, and/or GPR84. Contrary to octanoate and oleate, increased levels of Ghrl (orexigen; [31,32]) and decreased levels of Cck and Pyy (anorexigens; [31,32]) would suggest a stimulatory role in feed intake. Finally, the release of Ghrl and Glp-1 at a short time and of Cck and Pyy at a longer time would likely be responses occurring upon activation of gastrointestinal sensors of luminal PUFAs, such as ALA. In this case, hormonal release (especially at a short time) would suggest a contradictory effect on feed intake. It must be highlighted that the observation of hormonal release in response to ALA administration in rainbow trout indicates an important difference compared to mammals, in which n-3 PUFAs (such as ALA) do not seem to activate fatty acid sensors [40]. Future studies should focus on the determination of feed intake levels upon fatty acid intragastric administration to confirm changes in the abundance of gastrointestinal hormones observed in the present study.

In summary, the present study offers the first set of evidence supporting the presence of mechanisms able to sense fatty acids in the GIT lumen of rainbow trout. The data presented here show clear similarities to the widely accepted mammalian model of fatty acid gut sensing and its involvement in food intake regulation but also suggest several important differences (Figure 6). The most notorious of such differences is probably the lack within the rainbow trout genome of one of the main sensors of MCFAs and LCFAs in mammals, i.e., FFAR4, which appears to be compensated by other receptors binding and responding to these types of fatty acids. Another important difference lies in the ALA-induced modulation of fatty acid sensors and putative response triggered; this observation differs from mammals, in which no activation of fatty acid sensors seems to occur in response to n-3 PUFAs [41,42,43]. The differences between rainbow trout and mammalian fatty acid gut sensing mechanisms may be due to phylogenetical reasons (divergence between mammals and fish) and/or to the different dietary habits between carnivore (rainbow trout) and omnivore (mammalian models assessed so far) species of vertebrates. Further studies are required to study the basis for these differences.

## 4. Materials and Methods

### 4.1. Animals

Rainbow trout (body weight (bw) = 90 ± 20 g) were obtained from a local fish farm (A Estrada, Spain) and maintained in 100 L tanks (n = 40 fish/tank) with dechlorinated and aerated tap water (15 ± 1 °C) in an open circuit. The photoperiod was set to 12 h light:12 h darkness (12L:12D, lights on at 08:00 h). Fish were fed with a commercial dry pellet diet (proximate analysis: 44% crude protein, 21% crude fat, 2.5% carbohydrates, and 17% ash; 20.2 MJ kg^−1^ of feed; Biomar, Dueñas, Spain) daily at 11:00 until apparent visual satiety. All studies adhered to the ARRIVE Guidelines, were performed following guidelines of the European Union Council (2010/63/UE) and the Spanish Government (RD 53/2013) for the use of animals in research and were approved by the Ethics Committee of the Universidade de Vigo (00013-19JLSF).

### 4.2. Expression and Distribution of Fatty Acid Receptors and Transporters mRNAs along the Rainbow Trout Gastrointestinal Tract

Three 48 h-fasted fish were anesthetized in water containing 2-phenoxyethanol (0.02% *v*/*v*; Sigma-Aldrich, St. Louis, Missouri, USA) and euthanized by decapitation. Samples from the stomach, pyloric caeca, and intestine (proximal, anterior middle, intermediate middle, posterior middle, and distal; see Appendix A for graphical details) were collected, snap-frozen in dry ice and stored at −80 °C until quantification of the mRNA abundance of fatty acid receptors and transporters by RT-qPCR as described in Section 4.5. This experiment was repeated twice. Following RT-qPCR, representative samples of each tissue were run on 1.5% agarose gels, and single bands for each PCR were purified using QIAquick Gel Extraction Kit (Qiagen, Hilden, Germany) and sent for sequencing (CACTI, University of Vigo, Vigo, Spain). The specificity of the nucleotide-deduced sequences was analyzed using the BLAST tool (https://blast.ncbi.nlm.nih.gov/Blast.cgi?PROGRAM=blastn&PAGE_TYPE=BlastSearch&LINK_LOC=blasthome; accessed on 3 February 2023).

### 4.3. Characterization of the Response of Gastrointestinal Fatty Acid Sensing Mechanisms to the Luminal Presence of Fatty Acids

This experiment was performed on two consecutive days. For both days, fish scheduled for use in the experiment (maintained in acclimation tanks) fasted for 48 h so that intestinal emptying and basal levels of hormones involved in the metabolic control of food intake were achieved. On day 1, 30 fish were captured in batches of 6 (*n* = 6 per treatment) and slightly anesthetized with 2-phenoxyethanol (0.02% *v*/*v*). Then, intragastric administration of 1 mL. 100 g^−1^ bw of vehicle (distilled water containing 5% EtOH) alone (control) or containing 50 μmol.mL^−1^ of octanoate/octanoic acid (Sigma-Aldrich, Cat # C-2875), oleate/oleic acid (Sigma-Aldrich, Cat # O-1008), α-linolenate (ALA, Sigma-Aldrich, Cat # L2376) or sodium butyrate (Sigma-Aldrich, Cat # B5887) was performed. We selected octanoate, oleate, and ALA as representative MCFA, LCFA, and PUFA, respectively, because previous experiments from our research group demonstrated their effectiveness as feed intake modulators and/or modulators of related parameters in rainbow trout or Senegalese sole [44,45]. No available previous studies show a role for SCFAs in the control of feed intake in fish; butyrate was selected as representative in this study among the main SCFAs. To calculate the dose of fatty acid, we based on a typical amount of oleate (selected because we previously reported important effects of this fatty acid on feed intake in rainbow trout [44,46,47]) ingested daily by a trout fed with a standard commercial diet [48]). We then used an equimolar dose for the remaining fatty acids. Administration of treatments was carried out with a 13 cm-long cannula attached to a blunt-tip syringe. Putative regurgitation was checked visually, and we did not observe any during treatment administration. After intragastric treatments, fish from each experimental group were placed in individual tanks for recovery. After 20 min, they were again anesthetized to collect blood samples and, subsequently, plasma, which was used to determine the circulating levels of glucose, lactate, triglyceride, and free fatty acid (see Section 4.4). Then, fish were sacrificed by decapitation, and stomach and intestine (proximal, middle, and distal) samples were collected (see Appendix A for a graphical description of the regions sampled) for RT-qPCR or Western blot analysis; see below). We selected 20 min as sampling time based on preliminary experiments demonstrating this time to be adequate for a dye-containing saline solution to reach the middle/distal intestine after intragastric administration. On day 2, 30 fish per day were captured and intragastrically administered as described above, but sample collection was carried out 2 h post-administration.

### 4.4. Assessment of Plasma Metabolite Levels

Plasma levels of lactate, glucose, triglyceride, and free (non-esterified) fatty acid were assessed as indicators of the metabolic status of fish during experiments. Levels of all metabolites were assessed enzymatically using commercial kits adapted to a microplate format (For glucose, lactate, and triglyceride: Spinreact, Barcelona, Spain; for fatty acid: Fuji, Neuss, Germany). Results from these analyses are included in Appendix A. Levels of all parameters tested showed values comparable to those previously detected in healthy, unstressed individuals of the same species, with no considerable significant differences observed among groups, allowing us to consider that fish used for experiments have an adequate metabolic status and that fish were not exposed to major stress during experiments.

### 4.5. Quantification of mRNA Abundance by Reverse Transcription—Quantitative Polymerase Chain Reaction (RT-qPCR)

Isolation of total RNA from tissues and DNase treatment (*n* = 6 fish) were carried out using Trizol reagent (Life Technologies, Grand Island, Nebraska, USA) and RQ1-DNAse (Promega, Madison, Wisconsin, USA), respectively, as directed by the manufacturers. Optical density (OD) absorption ratio (OD 260 nm/280 nm) was used as an indicator of RNA purity, and it was determined using a NanoDrop 2000c (Thermo, Vantaa, Finland); only samples with an OD 260 nm/280 nm ratio > 1.8 were used for analysis. Following DNase treatment, 2 μg of total RNA was reverse transcribed into cDNA using Superscript II reverse transcriptase (Promega) and random hexamers (Promega) in a final volumen reaction of 20 μL, following manufacturer’s guidelines. Finally, using specific forward and reverse primers, mRNA abundance was quantified by RT-qPCR using MAXIMA SYBR Green qPCR Mastermix (Life Technologies). Specific primers to *ffar1*, *ffar2b1.1*, *ffar2b1.2*, *ffar2b2a*, *ffar2b2b*, *ffar2a1b,* and *ffar2a2* were designed based on rainbow trout cDNA sequences obtained in a previous study of our research group. Among the 10 *ffar* isoforms described in such a study to be present in the rainbow trout genome, we selected the 7 mentioned above because they are the most abundantly expressed in the trout intestine. Primers to *fatp4* were designed from the nucleotide sequence of *Salmo salar* (GenBank ID: XM_014138749.1) and positively checked for specificity within the rainbow trout genome using Genoscope (https://www.genoscope.cns.fr/trout/; accessed on 3 February 2023). Primers to *gpr84*, *gpr119*, *cd36,* and *slc16a1* (gene encoding Mct-1; two isoforms, *a* and *b*), as well as those to intracellular signaling elements and gastrointestinal hormones, were designed from rainbow trout nucleotide sequences available on GenBank, using Primer-BLAST online tool (https://www.ncbi.nlm.nih.gov/tools/primer-blast/; accessed on 3 February 2023). All primers used are included in Table 1 and were ordered from IDT (Leuven, Belgium). PCRs were performed in 96-well plates using 1 µL cDNA (replaced by water and RNA for controls) and 500 nm of forward and reverse primers in a final volume of 10 µL. Each sample was run in duplicate wells. All qPCRs were carried out in an iCycler iQ (Bio-Rad, Hercules, California, USA). Cycling conditions for qPCRs consisted of an initial step at 95 °C for 10 min, followed by 40 cycles at 95 °C for 30 s and 60 °C (except for *gcg* and *itpr3*, whose annealing temperature is 59 °C, and *ffar2b1.1* and *ffar2a1b*, with an annealing temperature of 62 °C; see Table 1) for 30 s. We included a melting curve (temperature gradient at 0.5 °C/5 s from 65–95 °C) at the end of each run to ensure that a single amplicon was being amplified. R^2^ of all reactions was 0.97–1, and efficiency was 95–100%. Following PCRs, resulting products were run on 1.5% agarose gels to confirm that a single product of the expected size was being amplified. The relative abundance of target transcripts was calculated using the 2-ΔΔCt method [49], using *actb* (gene encoding β-actin) and *ef1a* (gene encoding elongation factor 1α) as reference genes. These two genes were both stably expressed in this experiment.

### 4.6. Analysis of Protein Levels by Western Blot

Western blot analysis was performed from tissue samples from 6 fish. Extraction and quantification of protein were carried out as previously described [27]. Then, 50 µg protein was mixed with 4x Laemmli buffer containing 0.2% 2-mercaptoethanol (Bio-Rad) and denatured at 95 °C for 10 min. Then, samples were electrophoresed in Stain-Free 20% acrylamide gels (Bio-Rad) and transferred to a nitrocellulose membrane (0.2 µm pore-size; Bio-Rad) with the use of the Trans-Blot Turbo transfer system (Bio-Rad). After 60 min-blocking using Pierce Protein-Free T20 (PBS) Blocking Buffer (ThermoFisher), a specific primary antibody was added to the membrane and allowed to incubate overnight. Primary antibodies used for detecting gastrointestinal hormones in the stomach and intestine were custom synthesized as rabbit-raised polyclonal antibodies against synthetic peptide synthesized based on rainbow trout sequences (GenScript, Piscataway, NJ, USA). The exact antigen peptide sequences used are as follows: Ghrl: SQKPQVRQGKGKPPC (UniProtKB: Q76IQ4), Cck: CRPSHSQDEDKPEPP (UniProtKB: Q9YGE3), and Pyy: YPPKPENPGEDAPPC (UniProtKB:A0A060X2J5). All antibodies were diluted 1:500. After washing, membranes were incubated with secondary antibody (goat anti-rabbit IgG (H + L) HRP conjugate; Cat # ab205718, Abcam, Cambridge, United Kingdom) diluted to 1:5000. Clarity Western ECL substrate (Bio-Rad) was used to visualize proteins in a ChemiDoc Touch imaging system (Bio-Rad). We quantified protein bands by densitometry using Image Lab software and expressed results relative to the amount of total protein.

### 4.7. Statistical Analysis

All data were first checked for homogeneity of variance and normality, and, in case of failure of any of these requirements, they were log-transformed and re-assessed. Then, statistical differences among groups were assessed by one-way ANOVA followed by Dunnett’s test (for the in vivo intragastric experiment) or the Student-Newman-Keuls test (for plasma metabolite levels). Significance was considered when *p* < 0.05. SigmaPlot version 12.0 (Systat Software Inc., San Jose, CA, USA) statistical package was used to carry out all analyses.

## Figures and Tables

**Figure 1 ijms-24-04275-f001:**
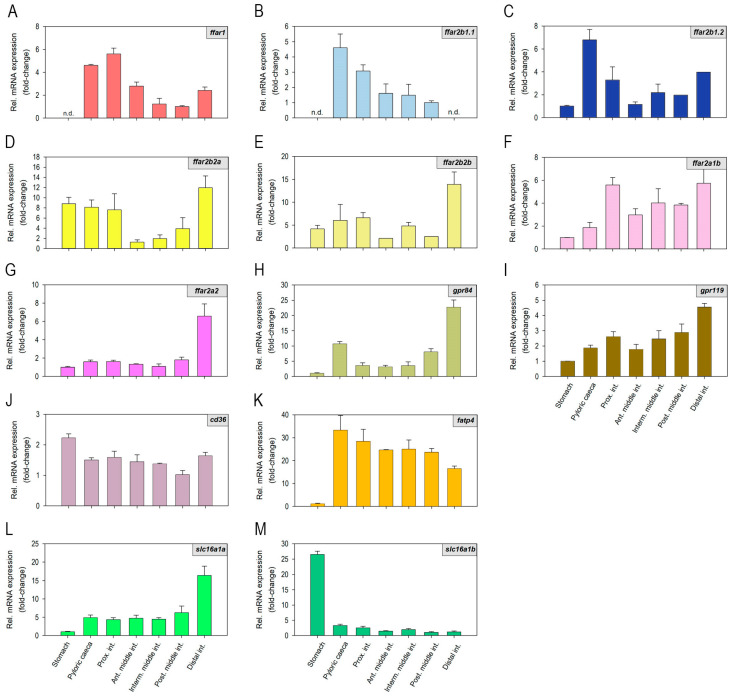
(**A**–**M**) Distribution of mRNAs encoding fatty acid receptors and transporters in the rainbow trout gastrointestinal tract. The abundance of mRNAs was quantified by RT-qPCR, with *actb* and *ef1a* considered reference genes. Data are expressed as mean + SEM (*n* = 6) relative to the tissue with the lowest mRNA abundance. Ant. Middle int., anterior middle intestine; *cd36*, cluster of differentiation 36 (gene encoding Fat/cd36); dist. Int., distal intestine; *fatp4*, fatty acid transporter 4; *ffar*, free fatty acid receptor; *gpr*, G protein coupled-receptor; interm. Middle int., intermediate middle intestine; post. Middle int., posterior middle intestine; prox. Int., proximal intestine; *slc16a1*, solute carrier family 16 member 1 (gene encoding Mct-1).

**Figure 2 ijms-24-04275-f002:**
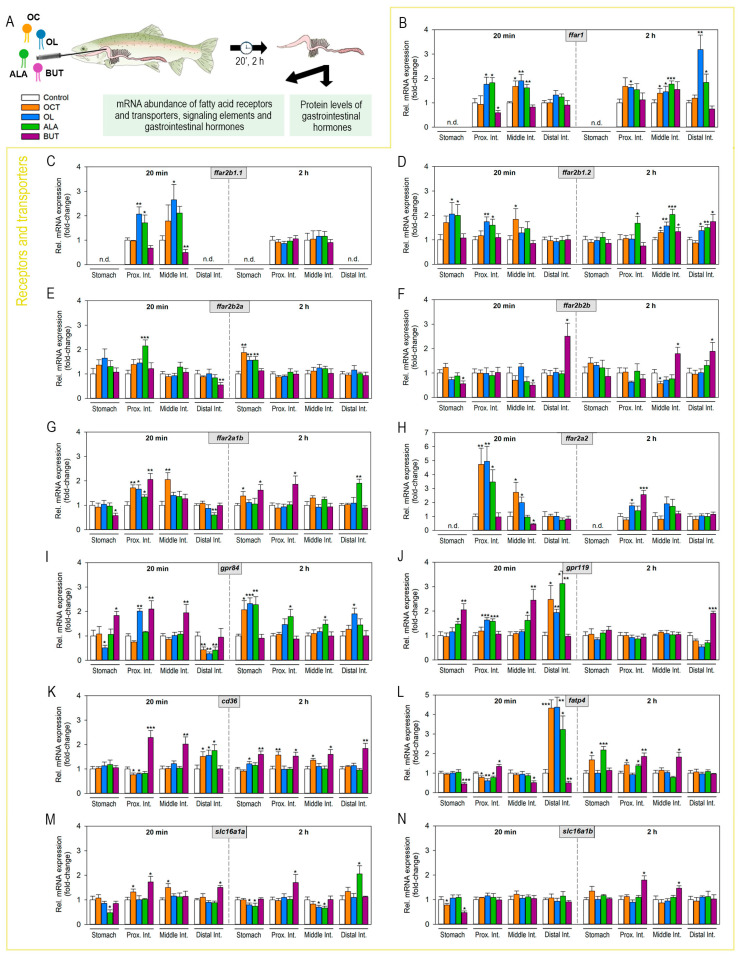
Effects of intragastrically administered octanoate, oleate, ALA, and butyrate on the expression of mRNAs encoding fatty acid receptors and transporters in the rainbow trout gastrointestinal tract. (**A**) 1 mL per 100 g^−1^ bw of vehicle (distilled water containing 5% EtOH) alone (control) or containing 50 μmol per mL^−1^ of octanoate, oleate, ALA, or sodium butyrate was intragastrically administered to rainbow trout. Stomach and intestine (proximal, middle, and distal) samples were collected at 20 min and 2 h post-treatment to assess several parameters related to fatty acid sensing and appetite regulation. Different results are shown in Figure 2, Figure 3 and Figure 4. (**B**–**N**) Abundance of mRNAs encoding fatty acid receptors/transporters in rainbow trout stomach and/or intestine (proximal, middle, and/or distal) 20 min and 2 h after intragastric administration of vehicle alone or containing octanoate, oleate, ALA or butyrate. The abundance of mRNAs was quantified by RT-qPCR, with *actb* and *ef1a* considered reference genes. Data are expressed as mean + SEM (*n* = 6) relative to the control group. Asterisks indicate statistical differences among groups (* *p* < 0.05, ** *p* < 0.01, *** *p* < 0.001), as assessed by one-way ANOVA followed by Dunnett’s testALA, α-linolenate; BUT, butyrate; *cd36*, cluster of differentiation 36 (gene encoding Fat/cd36); *fatp4*, fatty acid transporter 4; *ffar*, free fatty acid receptor; *gpr*, G protein coupled-receptor; *mct*, monocarboxylate transporter; OC, octanoate; OL, oleate.

**Figure 3 ijms-24-04275-f003:**
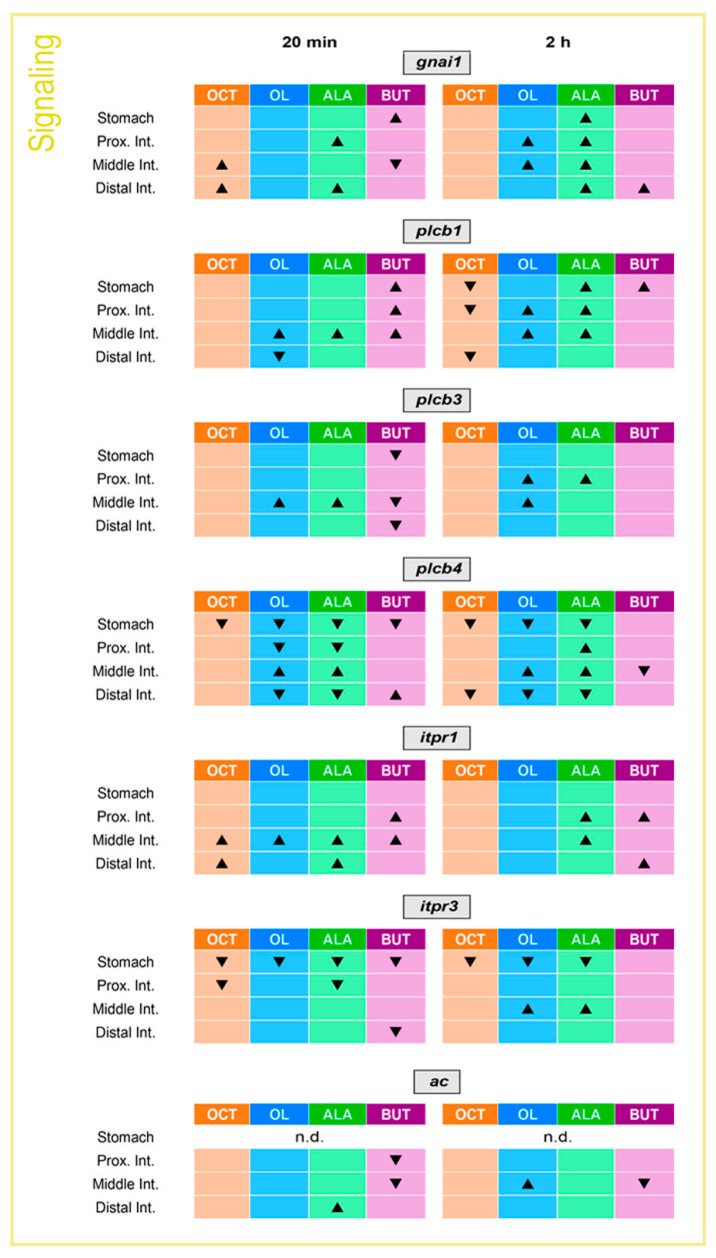
Effects of intragastrically administered octanoate, oleate, ALA, and butyrate on the abundance of mRNAs encoding intracellular signaling elements in the rainbow trout gastrointestinal tract. The abundance of mRNAs encoding intracellular signaling elements in the stomach and intestine (proximal, middle, and/or distal) of rainbow trout 20 min and 2 h following administration of vehicle alone or containing octanoate, oleate, ALA, or butyrate. The abundance of mRNAs was quantified by RT-qPCR, with *actb* and *ef1a* considered reference genes. In this figure, a table indicating significant changes in expression between treatment and control groups is included. ▲ indicates significant upregulation, ▼ significant downregulation, and the absence of any symbol indicates no significant variation in mRNA abundance. Please refer to Appendix A for values of mean ± SEM. *ac*, adenylate cyclase; ALA, α-linolenate; BUT, butyrate; *gnai1*, guanine nucleotide-binding protein G subunit alpha 1; *itpr*, inositol 1,4,5-trisphosphate receptor; OC, octanoate; OL, oleate; *plc,* phospholipase C.

**Figure 4 ijms-24-04275-f004:**
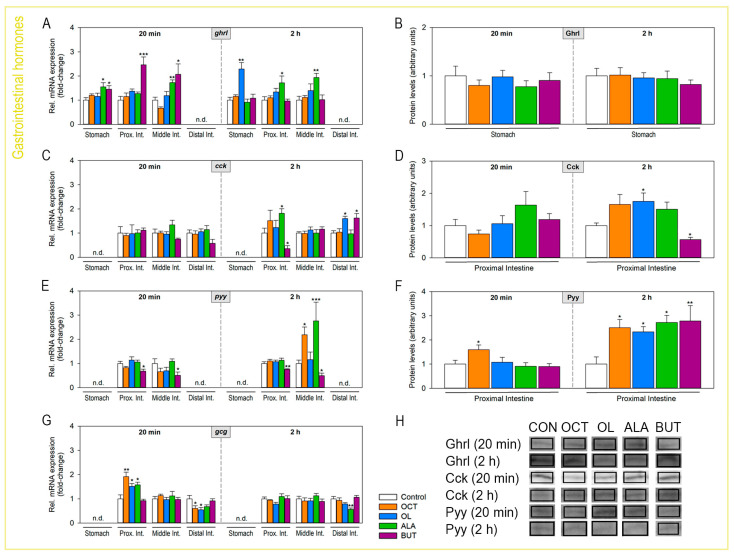
Effects of intragastrically administered octanoate, oleate, ALA, and butyrate on the expression of mRNAs encoding gastrointestinal hormones in the rainbow trout gastrointestinal tract. (**A**,**C**,**E**,**G**) The abundance of mRNAs encoding gastrointestinal hormones in rainbow trout stomach and/or intestine (proximal, middle, and/or distal) 20 min and 2 h after administration of vehicle alone or containing octanoate, oleate, ALA, or butyrate. Quantification was performed by RT-qPCR, with *actb* and *ee1a* considered reference genes. Data are shown as mean + SEM (*n* = 6) relative to the control group. A *t*-test was used to assess statistical differences between treatment and control groups. * *p* < 0.05, ** *p* < 0.01. (**B**,**D**,**F**,**H**) Protein abundance of gastrointestinal hormones in the stomach and intestine of rainbow trout 20 min and 2 h after intragastric administration of vehicle alone or containing octanoate, oleate, ALA, or butyrate. Data obtained by Western blot were normalized to the amount of total protein and are expressed as mean + SEM (*n* = 6) relative to the control group. Representative bands from each target protein are shown; please refer to Appendix A for entire blots. Asterisks indicate statistical differences among groups (* *p* < 0.05, ** *p* < 0.01, *** *p* < 0.001), as assessed by one-way ANOVA followed by Dunnett’s test. ALA, α-linolenate; BUT, butyrate; *cck*/Cck, cholecystokinin; *gcg*, proglucagon; *ghrl*/Ghrl, ghrelin; Glp-1, glucagon-like peptide-1; OC, octanoate; OL, oleate.

**Figure 5 ijms-24-04275-f005:**
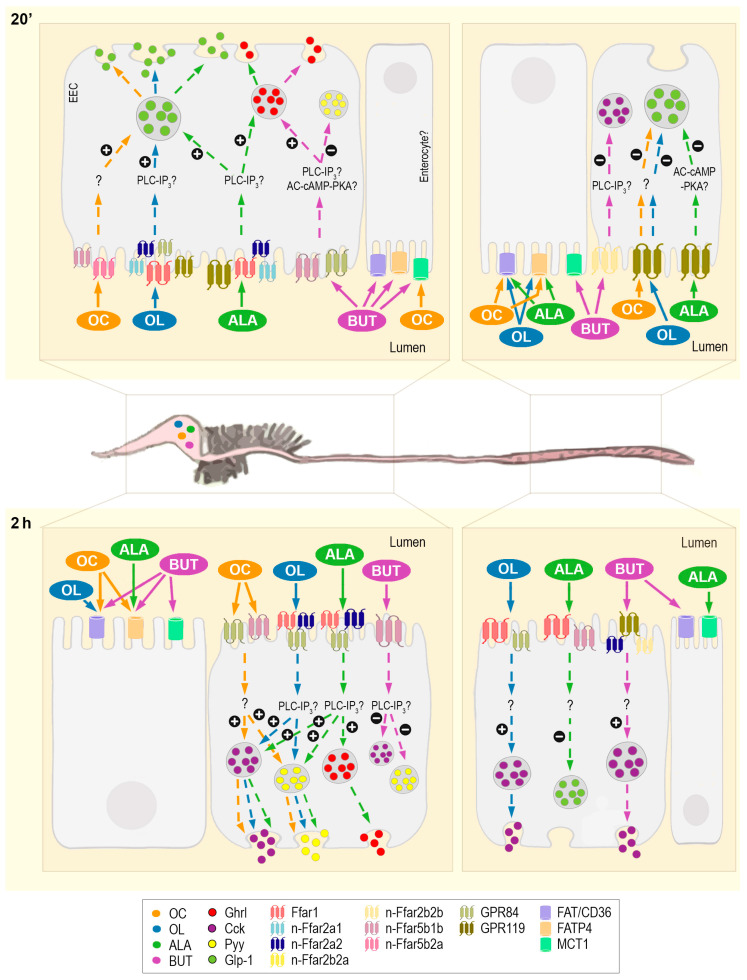
Schematic illustration showing the time- and region-dependent activation and the putative response of gastrointestinal fatty acid sensors to the luminal presence of fatty acids in rainbow trout. The proposed model for gut nutrient sensing in mammals is characterized by the sensing of luminal nutrients by specific receptors located in the apical face of EECs, the subsequent triggering of intracellular signaling pathways, and the ultimate basolateral release of gastrointestinal peptides (mainly ghrelin, CCK, PYY and GLP-1). In addition, specific transporters located in the brush border membrane of enterocytes are in charge of nutrient absorption. In the present research, evidence supporting such a nutrient-sensing model in the GIT of a fish species, the rainbow trout, was studied by the analysis of changes in the mRNA/protein levels of fatty acid receptors/transporters, intracellular signaling elements, and gastrointestinal hormones in response to intragastrically administered octanoate, oleate, ALA and butyrate. This illustration offers a graphic overview of the major results (in terms of positive changes) observed from such an experiment at the two time points analyzed (20 min and 2 h post-administration), showing the most likely sequence of events triggered as a consequence of the luminal presence of fatty acids. Results from the stomach, proximal intestine, and middle intestine were grouped, given their similar general tendency. Results from the distal intestine are presented separately. The distinction between the location of receptors in enteroendocrine cells and transporters in enterocytes is mostly based on the mammalian model; further experiments aimed at studying the specific location of both sensor types within the rainbow trout GIT are required to confirm such a distinction. AC-cAMP-PKA, adenylate cyclase—cAMP—protein kinase A intracellular signaling cascade; ALA, α-linolenate; BUT, butyrate; Cck, cholecystokinin; EEC, enteroendocrine cell; Fat/cd36, fatty acid transporter Cd36; Fatp4, fatty acid transporter 4; Ffar, free fatty acid receptor; Ghrl, ghrelin; Glp-1, glucagon-like peptide-1; Gpr, G protein coupled-receptor; Mct1, monocarboxylate transporter isoform 1; OC, octanoate; OL, oleate; PLC-IP_3_, phospholipase C—inositol triphosphate intracellular signaling cascase; Pyy, peptide tyrosine-tyrosine.

**Figure 6 ijms-24-04275-f006:**
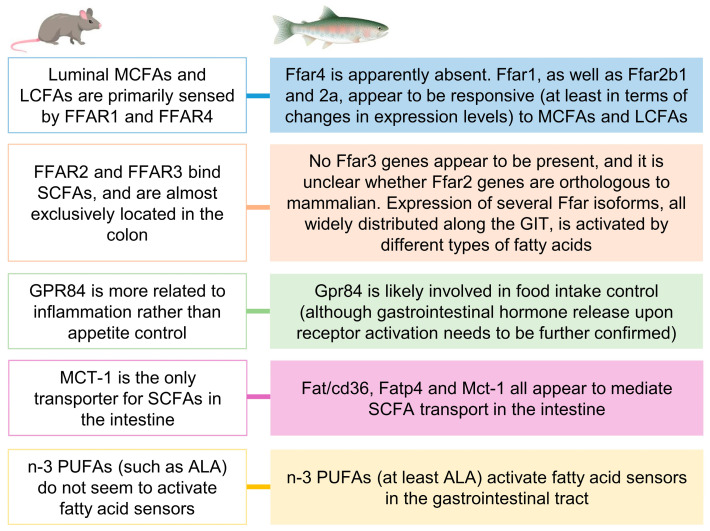
Major apparent differences in fatty acid gut sensing mechanisms between rainbow trout and the classical mammalian model, as suggested by results from the present research.

**Table 1 ijms-24-04275-t001:** Primers used for determining mRNA abundance in this study.

Gene	GenBank Accession Number	Sequence (5′ to 3′)	Reverse Primer (5′ to 3′)	Amplico-n Size (bp)	Annealing Temperature (°C)
*ac*	MF670431.1	CACCAGAAGTGTGCCAGCTA	GAGCAAACTCGGGTGGATCT	129	60
*actb*	NM 001124235.1	GATGGGCCAGAAAGACAGCTA	TCGTCCCAGTTGGTGACGAT	105	59
*ccka*	NM_001124345.1	GGGTCCCAGCCACAAGATAA	TGGATTTAGTGGTGGTGCGT	120	60
*cd36*	AY606034.1	CAAGTCAGCGACAAACCAGA	ACTTCTGAGCCTCCACAGGA	106	60
*ef1a*	AF498320	TCCTCTTGGTCGTTTCGCTG	ACCCGAGGGACATCCTGTG	159	59
*fatp4*	XM_014138749.1	GTAGCCTGGGAAACTTCGACA	TTCTTGCTGTTGGCTCCTTCG	244	60
*ffar1*	XM_036951038.1	CTGTGGTCATGCTGATGCTCT	CTTGGAAATGTTTGCTCCTGTC	188	60
*ffar2b1.1*	XM_021571760.2	CTTCCTCAGCGTGGCGTATC	CAGGTAGTGTTGTCGGCATCT	153	62
*ffar2b1.2*	XM_021571759.2	AGGCTGTTGATGACATGCACT	ATCTGATAGGGAAGGCCACA	147	60
*ffar2b2a*	XM_021561043.1	CACCTGAGCATTGTCGTCATC	TAATGAGCACGTTGGAGACGTTG	115	60
*ffar2b2b*	XM_021595167.2	ATGCCCTACTACAACCCACC	ACGTCACTAAGAGGCGCAATG	101	60
*ffar2a1b*	XM_021584265.2	CCTACCGCCAACTCAGCAAAC	AGTTCTCGTAGCAGACGGAG	147	62
*ffar2a2*	XM_021560940.2	CCCTTGTACGGAGTGGTGAG	CCAGCAGTGGCACGATGTAT	196	60
*gcg*	NM_001124698.1	AGGAGTGGTGCTCCATCCAAA	TCCTGATTTGAGCCAGGAAACA	111	59
*ghrl*	AB096919.1	GGTCCCCTTCACCAGGAAGAC	GGTGATGCCCATCTCAAAAGG	63	60
*gnai1*	CU073912	GCAAGACGTGCTGAGGACCA	ATGGCGGTGACTCCCTCAAA	150	60
*gpr84*	XM_021609929.1	GTTTTCGTGGGCTGTTTTGTC	CTGTTGAGCCAGGTGAGGTT	109	60
*gpr119*	NC_035086.1	TGAGATTGGCACCCGACTCT	CACAGAAGGAGTGGATGTTGGT	143	60
*itpr1*	XM_021569164.1	AGAAGAACGCCATGAGAGTGA	ACCACTTTGTCCCCTATCACC	121	60
*itpr3*	XM_021616029.1	GCAGGGGACCTGGACTATCCT	TCATGGGGCACACTTTGAAGA	64	59
*plcb1*	XM_036985415.1	GGAGTTGAAGCAGCAGAAGG	GGTGGTGTTTCCTGACCAAC	83	60
*plcb3*	XM_021577635.1	ATAGTGGACGGCATCGTAGC	TGTGTCAGCAGGAAGTCCAA	120	60
*plcb4*	XM_021600840.1	ACCTCTCTGCCATGGTCAAC	CGACATGTTGTGGTGGATGT	89	60
*pyy*	XM_021557532.1	GGCTCCCGAAGAGCTGGCCAAATA	CCTCCTGGGTGGACCTCTTTCCA	95	60
*slc16a1a*	XM_036947863.1	TGTTCGCCCGTCCTTCTATG	ACACAGGTAGGTCCACTGGT	347	60
*slc16a1b*	KF032405.1	CCACAGCCTGCAGTGAAAAGT	GCCAGAACAGACAGCAGGAAG	233	60

*ac, adenylate cyclase; actb*, β-actin; *ccka*, cholecystokinin a; *cd36*, cluster of differentiation 36 (gene encoding Fat/cd36); *ef1a*, elongation factor 1α; *fatp4*, fatty acid transporter 4; *ffar*, free fatty acid receptor; *gcg*, proglucagon (gene encoding Glp-1); *ghrl*, ghrelin; *gnai1*, guanine nucleotide-binding protein G subunit alpha 1; *gpr*, G protein coupled-receptor; *itpr*, inositol 1,4,5-trisphosphate receptor; *plcb*, phospholipase C-β; *pyy*, peptide tyrosine-tyrosine; *slc16a1*, solute carrier family 16 member 1 (gene encoding Mct-1).

## Data Availability

Data is available upon request to authors.

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
