# Peer review of "Fatty Acid Sensing in the Gastrointestinal Tract of Rainbow Trout: Different to Mammalian Model?"

_ijms, 2023, doi:10.3390/ijms24054275_

Round 1

Reviewer 1 Report

The paper entitled " Fatty acid sensing in the gastrointestinal tract of rainbow trout: evolutionary different to mammalian model?" by Calo and colleagues explores the presence of fatty acid receptors and transporters in the teleost gut and their response to fatty acids ingestion. The results show for the first time, expression of several fatty acid receptors and transporters in the fish gut. In addition, changes in the expression levels as a response to intragastric fatty acid loads suggest that they function as part of the fatty acid sensing mechanism as was described in mammals. The paper is well written and structured. The study is relevant, and the methodology employed is adequate for the proposed objectives. However, same aspects must be addressed before publishing the MS.

Major concerns

1) Figure 1. How many replicates have been used for quantification? The number of replicates could be included in the figure legends. Along with the bar graph, band of electrophoretic gels are shown. However, band intensity not always match with the bar height, probably due to biological variations of replicates. I wonder if this presentation helps to the readers or instead confuse them. Perhaps the hole gels can be included as supporting data. In panel K there are double bands, which was quantified? In panel Cand D a dye is obscuring the bands. it would be grate if these particular bands would be run with out dye.   

2) Authors inform that 20 min time point was choose because this was the time fore an aqueous dye to reach the intestine. However, how far into the intestine (proximal, middle or distal) the dye moves after 20 min? Would it be the same for fatty acid preparations. Furthermore, how 2 hs timepoint was selected? Is it expected that the FFAs still remain in the luminal surface of the gastrointestinal truck after 2 hs? Even in the anterior parts such as stomach? How the variations in concentration and localization would explain the changes in gene expression in the different gut sections?

3) Subsection 2.3. The objective was to analyze the second messengers involve after activation of fatty acid receptors. However, authors instead of measuring second messengers they measured the expression levels of same gene involved in the different pathways. Would not be possible to determine the second messenger pathways activation more directly, like measuring cAMP, IP3, enzymatic activities or phosphorylated substrates?

4) Statistical analysis. For the in vivo intra gastric approach authors used a t-test comparing each compound against the control group for each gut section and timepoint. However, I consider that a two-way anova followed by a pot-hoc test should be more adequate approach. Making several t-test instead is    a mistake.

Minor concerns

1) Title: The study is based on one teleost species, the rainbow trout, and the results are compared with the mammalian literature. There is not an evolutionary analysis, nor an evolutionary explanation of the differences found beyond species differences. Even as a question the term “evolutionary” sims out of place in the title.   

2) Supplementary figure 1. Which was the rational for selecting the bands?

3) Supplementary table 1. No significant differences have been indicated.

4) Supplementary table 2. No significant differences have been indicated. These results are not mentioned in result section nor in discussion section.

Author Response

We thank reviewer 1 for his/her nice words on our work. Concerns of the reviewer were addressed in the revised manuscript.

Reviewer 2 Report

The study in the article ‘Fatty acid sensing in the gastrointestinal tract of rainbow trout: evolutionary different to mammalian model?’aims to understand the expression and function of receptor and transport proteins involved in lipid sensing in the digestive system of rainbow trout.

Careful reading did not illuminate a significant need for major revisions. The text is well organized. The manuscript has a good experimental design and reporting. It supplements the study field with valuable information.

Minor:

The authors write that “ the receptor GPR119 has received attention as intestinal lipid sensor, although it does not bind FFAs directly but interacts with other endogenous lipid derivatives such as oleoylethanolamide (OEA) or dietary TG-derived 2-monoacylglycerols (2-MAG)’

This is not entirely true as GPR119 has been shown to be activated by FFAs such as palmitic and palmiotoleic acid https://doi.org/10.1016/j.nfs.2022.05.002

Author Response

We thank reviewer 2 for his/her nice and encouraging words on our work.

Round 2

Reviewer 1 Report

The paper has improved substantially, and the authors have addressed most of the concerns satisfactorily. However, the issue concerning the statistic analysis remains.

I can understand that the authors intention was to compare only the effects of each FFA against the control and not to compare between time points or among different FFAs. Even so, in the discussion section, authors compares both time points and the different FFA without a statistical support. Nevertheless, even accepting the objective to analyze only against the control for each FFA, using multiple t-test is not correct. From the statistical point of view, the variation have to be considered as a hole and in this case the tissue samples (gut section) are not independent since they correspond to the same fish. Consequently, the data treatment should be, at lease, though a one-way ANOVA for each FFA and each time point. From the viewpoint of data analysis, performing several t-test provoke that the type-1 (alfa) error to scale up. Thus, when you say "significantly different with p < 0.05" is simply not true. With 5 groups comparing all groups against each other the significance, instead of 0.05 (5%), increases up to 0.4 (40%). Comparing only against the control will imply a significance level around 0.25 (25%). This is the reason why performing several t-test should be avoided. When performing a post-hoc test this fact is internally considered and the individual significance is reduced accordingly to the Bonferroni’s correction in other to ensure a global significance level of 0.05  (Bland and Altman, 1995, BJM 310:170). Thus, for more that two groups it is mandatory to use ANOVA followed by a post-hoc test. Comparing only against the control group will increase the statistic power of the analysis, and you can use Dunnett's post-hoc test for that (see (https://statisticsbyjim.com/anova/post-hoc-tests-anova/) for a practical and rapid reference).    

Additionally, lines 234-237 seems redundant and in line 423 a stop is misplaced.

Author Response

The paper has improved substantially, and the authors have addressed most of the concerns satisfactorily. However, the issue concerning the statistic analysis remains.

We thank reviewer 1 for his/her nice words on our work. We have now changed the statistical analysis to fulfil reviewer’s concern.

I can understand that the authors intention was to compare only the effects of each FFA against the control and not to compare between time points or among different FFAs. Even so, in the discussion section, authors compares both time points and the different FFA without a statistical support. Nevertheless, even accepting the objective to analyze only against the control for each FFA, using multiple t-test is not correct. From the statistical point of view, the variation have to be considered as a hole and in this case the tissue samples (gut section) are not independent since they correspond to the same fish. Consequently, the data treatment should be, at lease, though a one-way ANOVA for each FFA and each time point. From the viewpoint of data analysis, performing several t-test provoke that the type-1 (alfa) error to scale up. Thus, when you say "significantly different with p < 0.05" is simply not true. With 5 groups comparing all groups against each other the significance, instead of 0.05 (5%), increases up to 0.4 (40%). Comparing only against the control will imply a significance level around 0.25 (25%). This is the reason why performing several t-test should be avoided. When performing a post-hoc test this fact is internally considered and the individual significance is reduced accordingly to the Bonferroni’s correction in other to ensure a global significance level of 0.05  (Bland and Altman, 1995, BJM 310:170). Thus, for more that two groups it is mandatory to use ANOVA followed by a post-hoc test. Comparing only against the control group will increase the statistic power of the analysis, and you can use Dunnett's post-hoc test for that (see (https://statisticsbyjim.com/anova/post-hoc-tests-anova/) for a practical and rapid reference).   

Following reviewer’s indication, results have now been statistically tested for significant differences using one-way ANOVA followed by Dunnett’s test.

Additionally, lines 234-237 seems redundant and in line 423 a stop is misplaced.

These have been changed in the revised manuscript.
